# Possible Role of miR-375-3p in Cervical Lymph Node Metastasis of Oral Squamous Cell Carcinoma

**DOI:** 10.3390/cancers16081492

**Published:** 2024-04-13

**Authors:** Masato Saika, Koh-ichi Nakashiro, Norihiko Tokuzen, Hiroyuki Shirai, Daisuke Uchida

**Affiliations:** Department of Oral and Maxillofacial Surgery, Graduate School of Medicine, Ehime University, Toon 791-0295, Ehime, Japan; saika.masato.ll@ehime-u.ac.jp (M.S.); tokuzen@m.ehime-u.ac.jp (N.T.); shirai.hiroyuki.li@ehime-u.ac.jp (H.S.); udai@m.ehime-u.ac.jp (D.U.)

**Keywords:** oral squamous cell carcinoma (OSCC), lymph node metastasis (LNM), microRNA-375-3p (miR-375), tumor suppressive-microRNA (TS-miR)

## Abstract

**Simple Summary:**

We focused on microRNAs (miRNAs) and investigated their usefulness as predictive markers for latent cervical lymph node metastasis (LNM) in early oral squamous cell carcinoma (OSCC). In miRNA microarray analysis, RT quantitative PCR, and digital PCR, the expression levels of miR-375-3p were significantly reduced in primary OSCC tissues with latent cervical LNM. Next, we examined the effects of miR-375-3p mimics on the growth and migration of four human OSCC cell lines that do not express miR-375-3p. When miR-375-3p mimics were introduced into human OSCC cells, cell proliferation and migration were significantly suppressed. Furthermore, a microarray and the Ingenuity Pathway Analysis miRNA target filter found 37 target gene candidates of miR-375-3p. Of these genes, the knockdown of CEPT1 and TIMM8A expression significantly inhibited the migration of human OSCC cells, similar to miR-375-3p mimics. The downregulation of miR-375-3p could promote cervical LNM by enhancing cell proliferation and migration in OSCC.

**Abstract:**

No clinically useful predictors of latent cervical lymph node metastasis (LNM) in early oral squamous cell carcinoma (OSCC) are available. In this study, we focused on the microRNAs (miRNAs) involved in the expression of numerous genes and explored those associated with latent cervical LNM in early OSCC (eOSCC). First, microarray and RT-PCR analyses revealed a significant downregulation of miR-375-3p expression in primary eOSCC tissues with latent cervical LNM. Next, we examined the effects of miR-375-3p mimics on the growth and migration of four human OSCC cell lines that do not express miR-375-3p. The overexpression of miR-375-3p significantly suppressed the cell proliferation and migration of human OSCC cells in vitro. Furthermore, miR-375-3p mimics markedly inhibited the subcutaneously xenografted human OSCC tumors. Finally, we found the genes involved in the PI3K-AKT pathway and cell migration as target gene candidates of miR-375-3p in human OSCC cells. These findings suggest that miR-375-3p functions as a tumor suppressive-miRNA in OSCC and may serve as a potential biomarker for the prediction of latent cervical LNM in eOSCC and a useful therapeutic target to suppress OSCC progression.

## 1. Introduction

According to statistics from the International Agency for Research on Cancer, 377,713 new cases of lip and oral cavity cancers and 177,757 deaths from these cancers were recorded worldwide in 2020 [1]. Oral squamous cell carcinoma (OSCC) is the most common malignant tumor of the lips and oral cavity [2]. Treatment strategies for OSCC are determined based on clinical staging according to the TNM classification of the Union for International Cancer Control [3]. Tumors less than 4 cm in size with a depth of invasion of less than 10 mm (T1/2) and no cervical lymph node metastasis (LNM) (N0) are classified as early OSCC (eOSCC, stage I/II). Treatment options for advanced OSCC (stage III/IV) can be broadly divided into three categories. The main treatments include surgery, chemotherapy, and radiation therapy. Furthermore, in addition to these conventional treatments, new options such as immunochemotherapy and photoimmunotherapy are available. If surgery is chosen for advanced OSCC, a neck dissection must be required in addition to removing the primary tumor. Although numerous treatments have currently been established for human malignancies, the standard management for eOSCC is intraoral tumor resection or brachytherapy, according to the National Comprehensive Cancer Network (NCCN). Because brachytherapy is not available in most hospitals, intraoral radical resection is generally performed. Although it has become easier to evaluate the presence or absence of LNM before surgery owing to technological developments, including ultrasonography (US), positron emission tomography–computed tomography (PET-CT), and contrast-enhanced CT (CECT), postoperative cervical LNM still occurs at a high 20–30% probability due to latent LNM that cannot be detected even by US, PET-CT, or CECT [4,5,6,7]. The 5-year survival rate of patients with latent LNM is significantly reduced. Therefore, latent LNM is the most important prognostic factor in eOSCC [8]. The global standard for managing N0 in the neck is elective neck dissection (END) to remove the supraomohyoid lymph nodes, and previous studies have shown its benefit to survival [9,10,11]. On the other hand, END has negative effects due to the postoperative dysfunction caused by overtreatment. Furthermore, END carries the risk of facial nerve mandibular branch and accessory nerve palsy, which may reduce the postoperative quality of life of patients [12]. Because approximately 80% of patients are overtreated, END should be performed selectively. Recently, sentinel lymph node biopsy (SLNB) has also been recommended as neck management for eOSCC by the NCCN [13,14]. SLNB is non-inferior to END in terms of survival rate but superior in terms of neck functionality [15]. However, SLNB is not covered by health insurance and cannot be used in clinical practice in Japan. Furthermore, mouse experiments have revealed that tumor immunity is not functional, and the anti-tumor effect of PD-1 and PD-L1 antibodies is lost when the tumor-draining lymph nodes are removed [16]. In the future, neoadjuvant or adjuvant therapy with immune checkpoint inhibitors may improve prognosis by preserving lymph nodes with no obvious metastasis. Therefore, exploring novel biomarkers for predicting latent LNM in eOSCC to avoid non-beneficial END is necessary.

Numerous previous studies have suggested that the ability to invade the lymphatic system is not determined by primary tumor size; rather, specific gene alternations and dysregulations drive the molecular processes responsible for LNM [17]. Research has revealed that latent cervical LNM could be predicted based on the expression levels of 102 genes in the primary tumor tissues of head and neck squamous cell carcinoma; in particular, the diagnostic accuracy of N0 cases was 100%, making it possible to avoid non-beneficial END [18]. Another study showed that it was possible to predict 24 out of 28 cases (85.7%) of latent LNM based on the expression levels of 696 genes in the primary tumor tissues of OSCC [19]. Furthermore, in the management of the neck in eOSCC cases, the first step is to predict latent LNM based on the gene expression profile of the primary tumor tissues. Next, a policy of performing SLNB in latent LNM-positive cases and neck dissection only in cases with histopathologically proven metastases was proposed. As a result, non-beneficial END was avoided in 72% of the cases. In addition, 94% of patients were reportedly able to receive appropriate treatment and avoid complications [20]. However, gene expression profiling is considerably complex, such that simple and useful predictors for latent LNM in eOSCC are yet to be identified. Therefore, in this study, we focused on the microRNAs (miRNAs) involved in the expression of several genes.

miRNAs are small endogenous single-stranded non-coding RNAs of 18–25 nucleotides in length that are responsible for the post-transcriptional regulation of gene expression in approximately 60% of human protein-coding genes [21]. miRNAs are reportedly involved in multiple diseases, including cancer and neurodegenerative and cardiovascular diseases. In particular, by regulating gene expression, miRNAs are greatly involved in biological processes such as cell proliferation, migration, infiltration, and apoptosis, which are essential for cancer development and progression [22]. In OSCC, several miRNAs have been reported to be associated with development and progression [22,23,24]. We have also reported several miRNAs that affect tumor growth in OSCC, including miR-361-3p and miR-1289 [25,26]. In addition, miRNAs are potential biomarkers owing to their up- and downregulation due to the nature of cancer. Therefore, this study aimed to explore novel biomarkers for predicting latent cervical LNM and examine their functions by quantifying the expression of miRNAs involved in migration and proliferation.

## 2. Materials and Methods

### 2.1. Patient Samples

Thirty primary OSCC tissue samples were collected from patients diagnosed with tongue eOSCC at the Department of Oral and Maxillofacial Surgery, Ehime University Hospital, between March 2006 and October 2020. Of these patients, 15 had latent cervical LNM, and the remaining patients without cervical LNM were followed up for at least 2 years after initial treatment. Patients with and without LNM did not significantly differ in terms of age, sex, T classification, and clinical stage, except for the differentiation-based histopathological grade according to the WHO classification (Appendix A). This study was approved by the Institutional Review Board of the Ehime University Hospital (1712017).

### 2.2. Cells and Cell Culture

We used four human OSCC cell lines (SAS-L1, HSC2, HSC3, and Ca9-22) and a human immortalized keratinocyte cell line (HaCaT) supplied by the RIKEN BioResource Research Center through the National Bio-Resource Project of the MEXT/AMED, Japan. All cell lines were maintained in DMEM (Fujifilm Wako, Osaka, Japan) supplemented with 10% FBS (Thermo Fisher Scientific, Waltham, MA, USA), 100 U/mL penicillin, and 100 μg/mL streptomycin (Fujifilm Wako), referred to as “complete medium”. Cells were maintained in an incubator with a humidified atmosphere of 95% air and 5% CO_2_ at 37 °C. The authenticity of these cells was verified using short tandem repeat profiling (Bex, Tokyo, Japan).

### 2.3. Total RNA Extraction

Total RNA was isolated by lysing cells and tissues after homogenization with the use of a TissueLyser (Qiagen, Hilden, Germany) and ISOGEN reagent (Nippon Gene, Tokyo, Japan), following to the manufacturer’s protocol. Briefly, tissues were obtained from biopsy or surgical materials, put in a sterile 2 mL tube with 0.5 mL of ISOGEN, and stored at −80 °C. The total RNA concentration was quantified using a Qubit Fluorometer (Thermo Fisher Scientific) and stored at −80 °C until use.

### 2.4. miRNA Microarray

For miRNA expression in OSCC tissues, 500 ng of total RNA was used to generate biotin-labeled miRNAs, using an Affymetrix FlashTag^TM^ Biotin HSR RNA Labeling Kit (Thermo Fisher Scientific), which were then hybridized to a GeneChip^TM^ miRNA4.1 Array Strip (Thermo Fisher Scientific) according to the manufacturer’s instructions. After washing and staining the array strips, the signal was developed and scanned using an Affymetrix GeneAtlas System (Thermo Fisher Scientific). The obtained CEL files were log2 transformed using Expression Console^TM^ Software v1.4 (Thermo Fisher Scientific) and then quantile normalized using GeneSpring GX 14.9.1 software (Agilent Technologies, Santa Clara, CA, USA). Subsequently, the microarray data were deposited in the Gene Expression Omnibus database (GEO; GSE241289) according to the minimum information about a microarray experiment (MIAME) guidelines.

### 2.5. RT-qPCR

To examine the expression levels of miRNAs, cDNA was synthesized from 10 ng of total RNA using a TaqMan^TM^ Advanced miRNA cDNA Synthesis Kit (Thermo Fisher Scientific), according to the manufacturer’s instructions. Subsequently, PCR amplification was conducted in a 20 µL final reaction volume containing 1 µL of cDNA, 10 µL of TaqMan^TM^ Universal Master Mix II without UNG, 1 µL of TaqMan^TM^ Advanced MicroRNA Assays (miR-375-3p, miR-6746-5p, and miR-191-5p), and 8 µL of RNase-free water. The thermal-cycling conditions were a PCR initial activation step of 50 °C for 20 min and then 95 °C for 15 min followed by 40 cycles of 94 °C for 15 s and 60 °C for 1 min using a ViiA 7 (Thermo Fisher Scientific). The relative expression levels of miRNAs were evaluated using the comparative threshold cycle (Ct) method with miR-191-5p as an internal control (ΔΔCt method).

### 2.6. Digital PCR

To quantify miRNA expression levels, PCR amplification was carried out in a 15 µL final reaction volume containing 1.5 µL of cDNA, 7.5 µL of QuantStudio™ 3D Digital PCR Master Mix v2 (Thermo Fisher Scientific), 0.75 µL of TaqMan™ Advanced MicroRNA Assays (miR-375-3p), and 5.25 µL of RNase-free water. We injected 14.5 µL of the mixtures into a QuantStudio™ 3D Digital PCR 20 K Chip Kit v2 using a QuantStudio™ 3D Digital PCR Chip Loader. The chip was placed in a thermal cycler (ProFlex^TM^ PCR System; Thermo Fisher Scientific); the initial PCR activation step was at 96 °C for 10 min, followed by an amplification step comprising 40 cycles at 98 °C for 30 s and 60 °C for 2 min. The signal was scanned using a QuantStudio™ 3D Digital PCR instrument (Thermo Fisher Scientific), and the number of copies were calculated using QuantStudio™ 3D AnalysisSuite™ Cloud Software v3.1 (Thermo Fisher Scientific).

### 2.7. Cell Growth Assay

We examined the effect of miRNA mimics or small interfering RNAs (siRNAs) on the growth of four human OSCC cell lines (SAS-L1, HSC2, HSC3, and Ca9-22) using a WST-8 Assay kit (Dojindo, Kumamoto, Japan). Briefly, cells (3 × 10^3^/well) were seeded into a 96-well plate (Corning Life Sciences, Corning, NY, USA) in complete medium with 20 nM of a miRIDIAN microRNA human hsa-miR-375 mimic, a miRIDIAN microRNA mimic negative control #1, 10 nM of ON-TARGETplus siRNAs (Appendix A) or ON-TARGETplus non-targeting control siRNAs (all from Dharmacon, Horizon Discovery, Cambridge, UK), and 0.3 µL of Lipofectamine RNAiMAX (Thermo Fisher Scientific) in a 100 µL final volume. After 72 h of reverse transfection, we added 10 µL of Cell Counting Kit-8 reagent (Dojindo) per well. Two hours later, measurements were taken at 450–620 nm using an SH-1300Lab microplate reader (Corona Electric, Hitachinaka, Japan).

### 2.8. Cell Migration Assay

We examined the effect of miRNA mimics or siRNAs on the migration of four human OSCC cells (SAS-L1, HSC2, HSC3, and Ca9-22) using a CytoSelect™ 24-well Cell Migration Assay kit (Cell Biolabs, San Diego, CA, USA). The assay contained polycarbonate membrane inserts (8 µm pore size) in a 24-well plate. Cells (1.0 × 10^6^) were seeded in a 60 mm dish (Corning Life Sciences) and treated with 0.3% Lipofectamine RNAiMAX and 20 nM of a miRIDIAN microRNA human hsa-miR-375 mimic, a miRIDIAN microRNA mimic negative control #1, and 10 nM of ON-TARGETplus siRNAs (Appendix A) or ON-TARGETplus non-targeting control siRNAs. After 24 h of reverse transfection, a polycarbonate membrane insert was placed on a 24-well plate, and then 1.0 × 10^6^ transfected cells were seeded on top of the insert in serum-free DMEM. Thereafter, DMEM containing 10% FBS was added to the lower chamber to obtain an FBS concentration gradient. Non-migratory cells were removed from the top of the membrane, the migratory cells were stained, and their stained areas were randomly selected (*n* = 6) and analyzed using an inverted fluorescence phase-contrast microscope (BZ-X810; Keyence, Osaka, Japan).

### 2.9. Xenograft Model

We examined the effect of miR-375-3p in vivo growth. SAS-L1 cells (1.0 × 10^6^) were seeded in a 100 mm dish (Corning Life Sciences) and treated with 0.3% Lipofectamine RNAiMAX and 20 nM of a miRIDIAN microRNA human hsa-miR-375 mimic or a miRID-IAN microRNA mimic negative control #1 (Dharmacon, Horizon Discovery). After 24 h of reverse transfection, the cells were harvested and injected subcutaneously at two sites in the flanks of 5-week-old male Balb/c athymic nude mice (CLEA Japan, Tokyo, Japan) at a density of 1.5 × 10^6^ treated cells per 150 μL of plain DMEM. Tumor diameter was measured every three days starting one week after injection using digital calipers, and tumor volume (mm^3^) was calculated using the following formula: length × width × height × 0.523. Twenty-two days after injection, the xenografts were dissected and we measured the tumor weight by using an electronic balance (ATX224, Shimadzu, Kyoto, Japan).

### 2.10. Microarray and Pathway Analysis

We examined the molecular functions of miR-375-3p in four human OSCC cell lines (SAS-L1, HSC2, HSC3, and Ca9-22) with the use of a microarray and Ingenuity Pathway Analysis (IPA, Qiagen). Cells (3.0 × 10^5^) were seeded in 60 mm dishes and supplemented with 0.3% Lipofectamine RNAiMAX and 20 nM of an miRIDIAN microRNA human hsa-miR-375 mimic or an miRIDIAN microRNA mimic negative control #1 (Dharmacon, Horizon Discovery). After 48 h, total RNA was extracted from the cells. Subsequently, we used 100 ng of total RNA to generate biotin-labeled cRNA using the Affymetrix GeneChipTM 3′ IVT PLUS Reagent Kit (Thermo Fisher Scientific), and then cRNA was hybridized to the Affymetrix Human Genome U-219 Array Strips (Thermo Fisher Scientific) according to the manufacturer’s instructions. After washing and staining the array strips, scanning was performed using an Affymetrix GeneAtlas System (Thermo Fisher Scientific). Data analysis was performed by using GeneSpring GX 14.9.1 software (Agilent Technologies). The robust multichip average method, which uses background correction and normalization, was used. The data were deposited in the GEO database (GSE240389) according to the MIAME guidelines. Furthermore, the results were analyzed using IPA core analysis and a microRNA target filter (Qiagen). To identify potential target genes associated with the studied phenotype, Gene Set Enrichment Analysis (GSEA) was conducted with the array data. A result was considered statistically significant if the false discovery rate (FDR) < 25% (<0.25) or a *p*-value < 0.05.

### 2.11. Statistics

All in vitro experiments were performed in triplicate and repeated thrice. ANOVA was used to test the differences between three or more groups, and Student’s *t*-test was used to test the differences between two groups. Statistical analyses were performed using GraphPad Prism 9.5 (GraphPad Software, San Diego, CA, USA). *p* < 0.05 was considered significant. All data are presented as the mean ± standard deviation (SD); SD is indicated by error bars in the graphs.

## 3. Results

### 3.1. Identification of miRNAs Related to Latent Cervical LNM in Primary eOSCC Tissues

First, we attempted to identify miRNAs whose expression was significantly altered in relation to cervical LNM in primary eOSCC tissues using a miRNA microarray. Using GeneChip miRNA 4.1 Array Strips, we found 82 miRNAs that were expressed in primary eOSCC tissues with a more than two-fold change compared with the adjacent normal oral mucosa. Of these miRNAs, 33 were significantly upregulated (Appendix A), and 49 were downregulated (Appendix A). Furthermore, among these 82 miRNAs, the downregulation of miR-375-3p and miR-6746-5p was associated with latent LNM (Figure 1A). The expression levels of miR-375-3p were lowest in primary eOSCC tissues with latent LNM (Figure 1B).

Next, using RT-qPCR, we quantified the expression levels of miR-375-3p and miR-6746-5p in primary OSCC tissues. Similar to the results of the miRNA microarray analysis, miR-375-3p was significantly downregulated in primary eOSCC tissues with latent cervical LNM (Figure 1C). In contrast, the expression of miR-6746-5p was not detected in most primary eOSCC tissues using RT-qPCR. Subsequently, the expression levels of miR-375-3p in primary eOSCC tissues were quantified using dPCR. The dPCR results showed a more significant difference relative to the results obtained by RT-qPCR (Figure 1D). When the reference value for predicting latent cervical LNM was set to less than 58.0 copies based on the receiver operating characteristic curve, the sensitivity, specificity, accuracy, and area under the curve (AUC) were 80, 100, and 90%, and 0.942, respectively (Figure 1E). These results suggest that the expression level of miR-375-3p is a useful predictor of latent cervical LNM in eOSCC.

### 3.2. Expression and Function of miR-375-3p in Human OSCC Cells In Vitro

We examined the expression of miR-375-3p in OSCC cells. The expression levels of miR-375-3p in the four human OSCC cell lines (SAS-L1, HSC2, HSC3, and Ca9-22) and HaCaT cells were quantified using RT-qPCR. High miR-375-3p expression was observed in HaCaT cells, which represented normal epithelial cells. In contrast, miR-375-3p expression was not detected in any of the tested human OSCC cell lines (Figure 2).

To elucidate the function of miR-375-3p in the growth of human OSCC cells, we introduced an miR-375-3p mimic into SAS-L1, HSC2, HSC3, and Ca9-22 cells that did not express miR-375-3p. The WST-8 assay was used to evaluate the proliferation of OSCC cells. In all human OSCC cell lines, a significant growth inhibitory effect was observed in the miRNA-375-3p mimic-transfected and overexpressing cells compared with that in the non-targeting miRNA mimic (miR-NT) used as a negative control (Figure 3).

Subsequently, we examined the effect of the miR-375-3p mimic on the migration of OSCC cells using a transwell chamber assay. miR-373-3p overexpression significantly suppressed the migration of SAS-L1, HSC2, HSC3, and Ca9-22 cells by 97.3, 92.7, 51.4, and 92.5%, respectively (Figure 4A,B).

### 3.3. Effect of miR-375-3p Mimic on In Vivo Growth

The effect of miR-375-3p mimics on the in vivo growth of human OSCC cells was investigated in a mouse xenograft model. SAS-L1 cells were chosen for the in vivo assay due to their stable tumorigenicity compared to the other OSCC cells utilized. The study revealed a significant reduction in the size and weight of subcutaneously xenografted SAS-L1 tumors in the miR-375-3p mimic group compared to the control group (Figure 5). Throughout the administration of miR-375-3p mimics, there were no observed reductions in food intake or body weight in the mice.

### 3.4. Target Gene Candidates and Pathways of miR-375-3p in Human OSCC Cells

To elucidate the molecular mechanisms underlying the growth- and migration-inhibitory effects of miR-375-3p in human OSCC cells, we identified its target genes and pathways using microarray analysis, the IPA microRNA target filter, and IPA core analysis (Figure 6A). The analyses were performed with the total RNA that was extracted from the four human OSCC cell lines after transfection with miR-375-3p or the miR-NT mimic for 48 h. The expression levels of 266 genes were downregulated by miR-375-3p overexpression in all human OSCC cell lines tested (Appendix A). The IPA microRNA target filter showed that 37 genes had the target sequences of miR-375-3p in their 3′-UTR (Appendix A). Furthermore, IPA core analysis using these target genes indicated that miR-375-3p overexpression suppressed the expression of genes involved in activating the PI3K-AKT pathway (Figure 6B). Subsequently, we investigated whether the differentially expressed genes between the two groups, miR-NT and miR-375-3p, were biased toward a specific set of genes using GSEA. Unfortunately, GSEA did not show any changes in the expression of genes typically associated with LNM (Appendix A).

### 3.5. Identification of the miR-375-3p Target Gene Candidates Involved in the Growth and Migration of Human OSCC Cells

Using an siRNA knockdown library, we examined the target gene candidates of miR-375-3p that suppress the growth and migration of human OSCC cells. In SAS-L1 cells, the knockdown of three genes (PLEKHA3, POC1B, and TMEM55A) showed significant growth inhibitory effects (Figure 7A). Similar results were obtained for SAS-L1 and HSC2 cells; however, no effect was observed in HSC3 or Ca9-22 cells (Figure 7B).

On the other hand, the knockdown of 17 genes (QKI, UBE2E2, LSM12, PLEKHA3, CORO2A, PTPMT1, CSTF2, EIF4G3, PPPDE2, RPN1, MBD2, PRDX1, KIAA1524, MOBKL1B, CNN3, CEPT1, and TIMM8A) significantly suppressed SAS-L1 cell migration (Figure 8A). Furthermore, the suppression of CEPT1 and TIMM8A expression significantly inhibited the migration of all other human OSCC cell lines, including HSC2, HSC3, and Ca9-22 (Figure 8B).

## 4. Discussion

Although the efficacy of END has been established, approximately 80% of patients may be overtreated. We believe that END should be performed more selectively based on gene expression profiles in primary OSCC tissues. Therefore, we investigated the molecules involved in latent cervical LNM in primary eOSCC tissues. Here, we focused on miRNAs associated with the development and progression of human malignancies. Several miRNAs have been reported to function as oncogenes or tumor suppressor genes in OSCC [22,23,24,25,26]. In the present study, we showed that the downregulation of miR-375-3p expression was associated with latent cervical LNM of eOSCC, and the replenishment of miR-375-3p suppressed the growth and migration of human OSCC cells.

miR-375-3p is known to act as a tumor-suppressive miRNA (TS-miRNA) in numerous types of malignant tumors, including gastrointestinal stromal tumors (GISTs), colorectal cancer (CRC), hepatocellular carcinoma, esophageal squamous cell carcinoma (ESCC), laryngeal squamous cell carcinoma (LSCC), and breast cancer (BC) [27,28,29,30,31]. In LSCC, miR-375-3p was among the downregulated miRNAs found when comparing tumor and normal tissues [30]. In CRC, miR-375-3p binds to the target gene SP1 and inhibits cell proliferation [27,32]. The expression of miR-375-3p is downregulated in ESCC tissues, and its ectopic expression significantly inhibits cell migration and invasion, suggesting that miR-375-3p functions as an anti-metastatic miRNA in ESCC cells. Furthermore, in ESCC, miR-375-3p reportedly suppresses metastasis by targeting MMP13 [31]. miR-375-3p overexpression reduces the proliferation and migration of GIST and gastric cancer cells via the AKT/mTOR signaling pathways [28,33]. Our results are consistent with those of previous reports.

In particular, miR-375-3p is reportedly associated with epithelial–mesenchymal transition (EMT). The regulation of EMT comprises a complex network that includes signal transduction pathways typical of cancer metastasis, such as TGF-β, Wnt, Notch, and Smad signaling [27]. The downregulation of miR-375-3p expression promotes EMT in human gastric cancer cells [27,34]. In CRC, miR-375-3p targets SP1 to inhibit MMP2, vimentin, snail, β-catenin, and N-cadherin [27,32]. Therefore, in recent years, miR-375-3p has attracted attention as a therapeutic target and biomarker of several types of malignant tumors. Decreased miR-375-3p expression has been found to be associated with LNM and staging in liver cancer and is expected to be a biomarker for diagnosis [27,35]. In addition, it can be used as a biomarker for predicting the prognosis of patients with ESCC and is reportedly directly proportional to patient survival [27,36].

miR-375-3p has also been reported in head and neck cancers, including OSCC, and similar to our results, decreased expression was observed in tumor areas relative to potentially malignant oral lesions or normal oral epithelial areas [37,38,39]. In OSCC, IGF1R, SLC7A11, and KLF5 are reportedly miR-375-3p target genes and are involved in cell proliferation, migration, and invasion [37,38,40,41]. The detection of miR-375-3p in saliva may be useful as a biomarker for potential malignant diseases of the oral cavity [42]. However, no reports on the association of miR-375-3p with latent cervical LNM in eOSCC are available. To our knowledge, we are the first to report the possibility that the downregulation of miR-375-3p expression in primary eOSCC tissues is associated with latent cervical LNM. Nevertheless, further external validation is required, and eOSCC cases should be prospectively collected. In fact, we are prospectively collecting samples from patients with new eOSCC, including areas other than the tongue, and observing their progress.

Through in vitro and in vivo study, we showed that miR-375-3p overexpression inhibited the growth and migration of human OSCC cells. Furthermore, we identified novel target gene candidates and examined their functions using microarray analysis, IPA, and siRNA knockdown libraries. Among the miR-375-3p target gene candidates, CEPT1 and TIMM8A knockdown suppressed the migration of all human OSCC cell lines tested in the present study. However, the role of CEPT1 in oncology is yet to be elucidated. Furthermore, TIMM8A is an oncogene whose upregulation is correlated with poor prognosis in BC [43,44]. However, TIMM8A in OSCC has not been reported. Here, we found that CEPT1 and TIMM8A are novel target gene candidates of miR-375-3p that regulate human OSCC cell migration. However, this study has limitations in that RNA immunoprecipitation (RIP) and luciferase assays were not performed. In order to identify these genes as direct target genes of miR-375-3p, we plan to conduct further in vitro assays.

In vivo studies also showed the inhibitory effect of an miR-375-3p mimic on OSCC tumor growth. Although there were reports of cervical LNM models in nude mice, reproducibility was difficult because cervical LNM did not occur stably. In addition, this study focused on eOSCC, and when conducting research related to latent cervical LNM, it is difficult to use the animal models due to starvation and death by tumor growth within two weeks. Although we understand the importance of migration, invasion, EMT), etc., in LNM, it is believed that tumor expansion or growth at the primary site also contribute to LNM due to the tumor’s proximity to lymph flow. Therefore, we here tested the effect of an miR-375-3p mimic on the in vivo growth using a stable mouse model in our laboratory. These results showed that miR-375-3p markedly inhibited tumor growth and is likely to be involved in LNM in OSCC.

## 5. Conclusions

The downregulation of miR-375-3p expression was associated with latent cervical LNM in primary eOSCC tissues. Subsequently, the overexpression of miR-375 suppressed the growth and migration of human OSCC cells. The expression levels of miR-375-3p in primary tumor tissues appear to be a useful biomarker for predicting latent cervical LNM in eOSCC. Furthermore, miR-375-3p functions as a tumor-suppressive miRNA and may be a useful therapeutic target in OSCC.

## Figures and Tables

**Figure 1 cancers-16-01492-f001:**
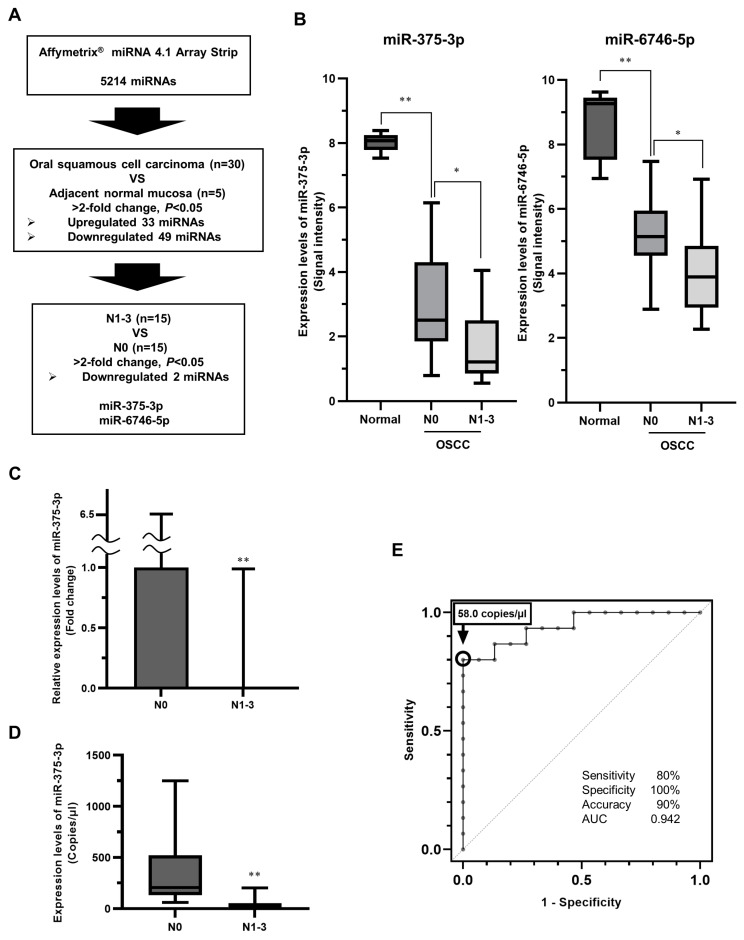
Identification of miRNAs associated with latent cervical LNM in primary eOSCC tissues. (**A**) A comprehensive expression analysis of miRNAs using microarrays identified 82 miRNAs with significantly higher than 2-fold expression changes in primary OSCC tissues compared with the adjacent normal mucosa. Furthermore, among these miRNAs, the decreased expression of miR-375-3p and miR-6746-5p was significantly associated with latent cervical LNM. (**B**) The signal intensity of miR-375-3p and miR-6746-5p expression in the miRNA4.1 array. miR-375-3p expression was significantly downregulated in primary OSCC tissues with latent cervical LNM. (**C**) miR-375-3p expression in primary OSCC tissues determined by RT-qPCR. The miR-375-3p expression level was significantly reduced in OSCC with LNM. (**D**) miR-375-3p expression in primary OSCC tissues determined by digital PCR. Comparable results were obtained with RT-qPCR. (**E**) The prediction of latent cervical LNM by the copy number of miR-375-3p. When the reference value for predicting latent cervical LNM was less than 58.0 copies based on the receiver operating characteristic curve, the sensitivity, specificity, accuracy, and area under the curve (AUC) were 80, 100, and 90%, and 0.942, respectively. * *p* < 0.05 and ** *p* < 0.01 compared with the control. miRNA, microRNA; LNM, lymph node metastasis; eOSCC, early oral squamous cell carcinoma.

**Figure 2 cancers-16-01492-f002:**
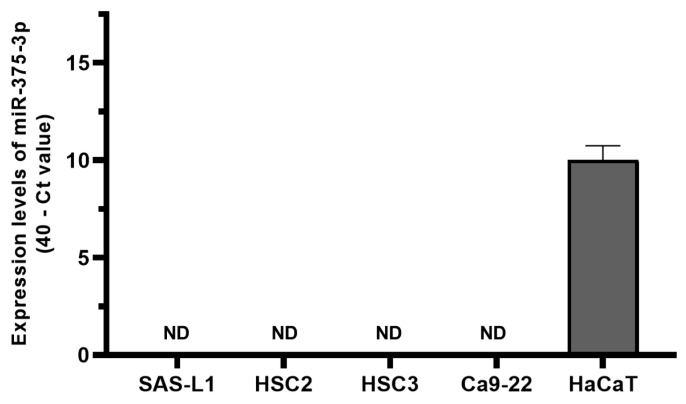
The expression levels of miR-375-3p in human OSCC cells. Four human OSCC cell lines (SAS-L1, HSC2, HSC3, and Ca9-22) and the human immortalized keratinocyte cell line HaCaT were used. The expression of miR-375-3p was not detected in SAS-L1, HSC2, HSC3, and Ca9-22 cells by RT-qPCR. ND, not detected; OSCC, oral squamous cell carcinoma.

**Figure 3 cancers-16-01492-f003:**
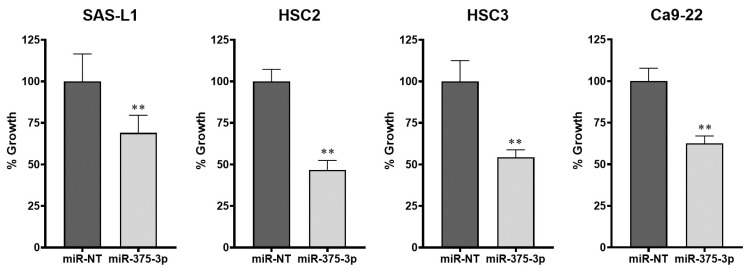
The effect of an miR-375-3p mimic on the growth of human OSCC cells. Reverse transfection was performed for 72 h to examine the growth inhibitory effect of an miR-375-3p mimic on human OSCC cells. miR-375-3p overexpression significantly inhibited the proliferation of all human OSCC cell lines. miR-NT, miR-non-target; ** *p* < 0.01 compared with the control; OSCC, oral squamous cell carcinoma.

**Figure 4 cancers-16-01492-f004:**
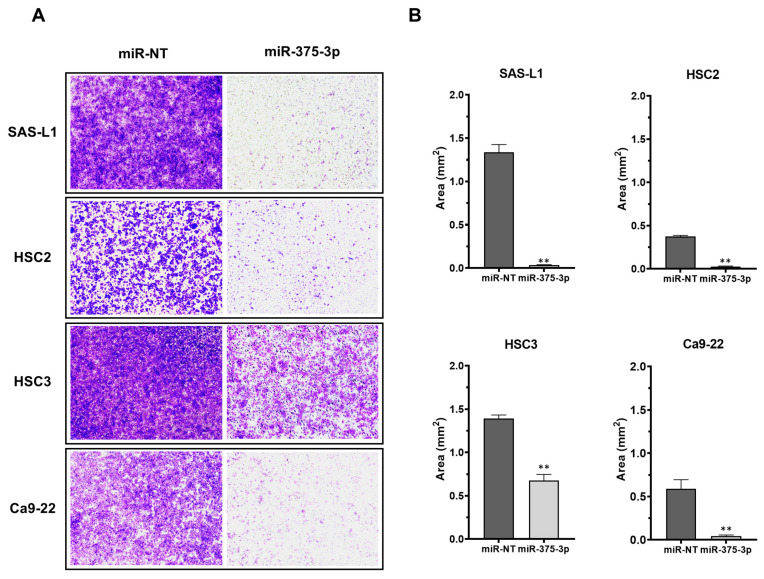
The effect of an miR-375-3p mimic on the migration of human OSCC cells. (**A**) Migratory cells at the bottom of the polycarbonate membrane were stained (×4). (**B**) miR-375-3p overexpression significantly suppressed the migration of all human OSCC cell lines. miR-NT, miR-non target; **, *p* < 0.01 compared with the control. OSCC, oral squamous cell carcinoma.

**Figure 5 cancers-16-01492-f005:**
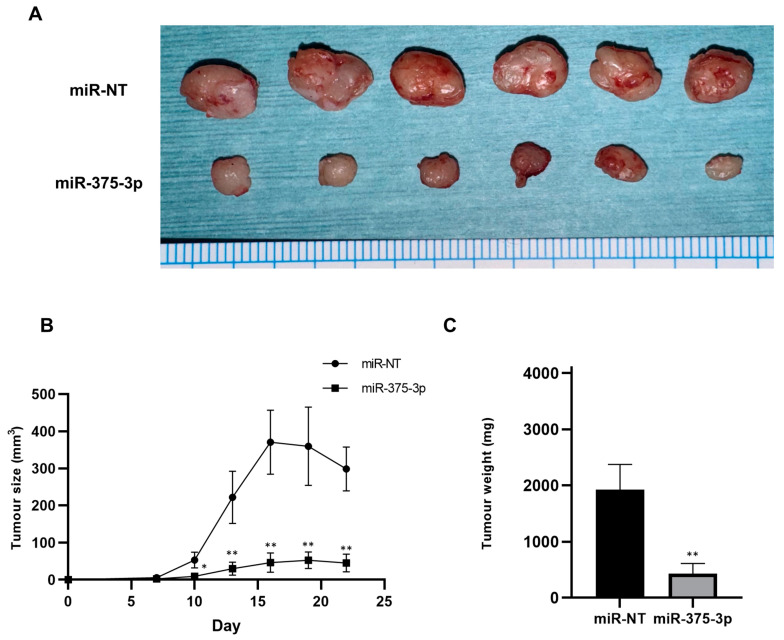
In vivo tumor growth after miR-375-3p mimic introduction. (**A**) Images of tumors that were sacrificed and excised 22 days after the injection of the miR-375-3p mimic. (**B**) Tumor volume was measured every 3 days from 1 week after tumor injection in the mice. (**C**) Tumor weight at 22 days after the first administration of the miR-375-3p mimic. * *p* < 0.05 and ** *p* < 0.01 compared to the control group. miR-NT, miR-non-target.

**Figure 6 cancers-16-01492-f006:**
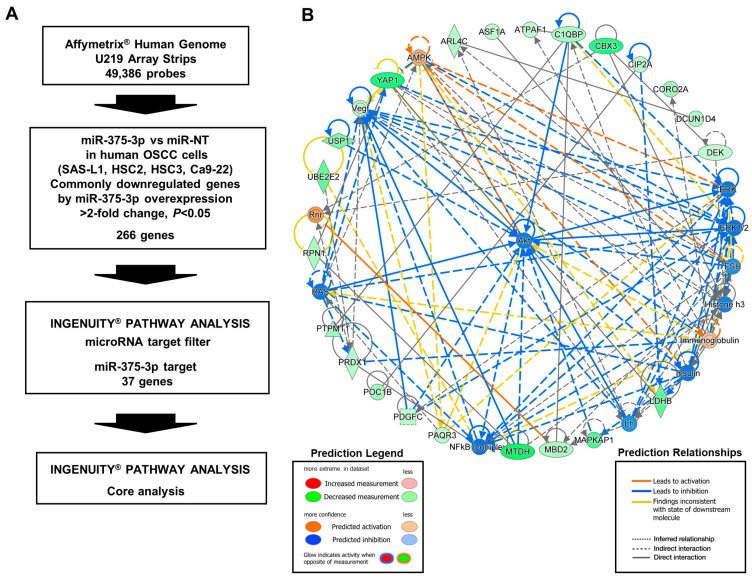
Target gene candidates and pathways of miR-375-3p in human OSCC cells. (**A**) miR-375-3p or miR-NT mimics were introduced into SAS-L1, HSC2, HSC3, and Ca9-22 cells. In microarray analysis, miR-375-3p overexpression significantly reduced the expression of 266 genes. Of these genes, the IPA microRNA target filter showed that 37 genes had the target sequences of miR-375-3p in their 3′-UTR. (**B**) IPA core analysis showed that miR-375-3p overexpression suppressed the expression of genes involved in the activation of PI3K-Akt pathways. miR-NT, miR-non-target; OSCC, oral squamous cell carcinoma; IPA, Ingenuity Pathway Analysis.

**Figure 7 cancers-16-01492-f007:**
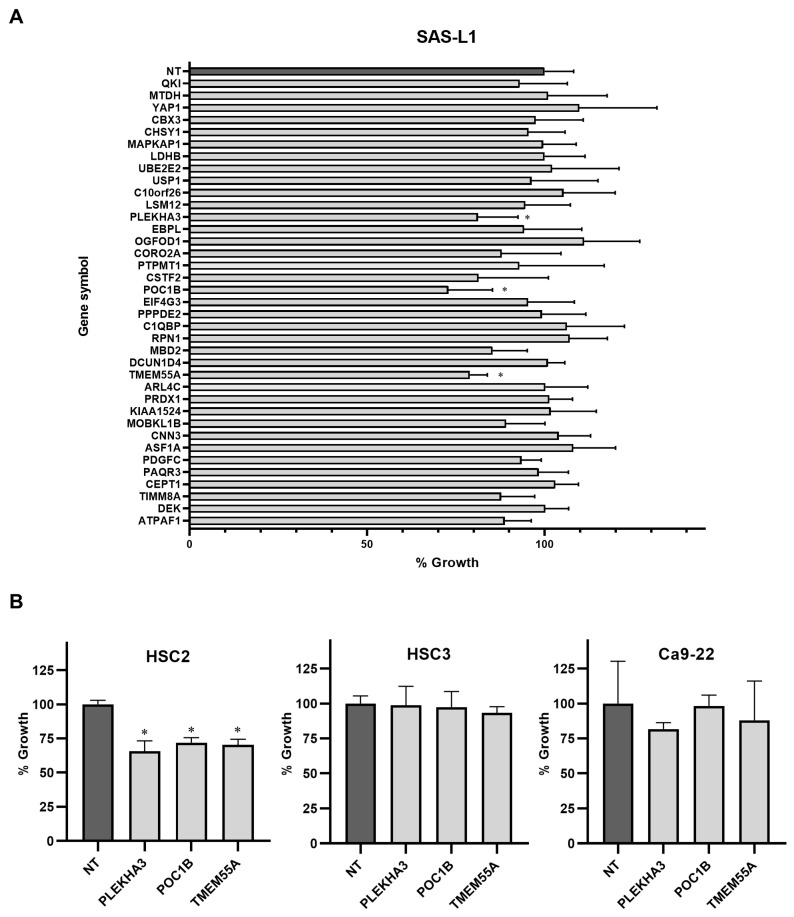
Identification of target genes involved in the growth inhibitory effect of miR-375-3p. (**A**) When we introduced siRNAs specific to miR-375-3p target genes into SAS-L1 cells, PLEKHA3, POC1B, and TMEM55A knockdown significantly suppressed the growth of these cells. (**B**) No genes were commonly involved in the growth inhibitory effect of miR-375-3p on all human OSCC cell lines. NT, non-targeting control siRNA; * *p* < 0.05 compared with the control; OSCC, oral squamous cell carcinoma; siRNA, small interfering RNA.

**Figure 8 cancers-16-01492-f008:**
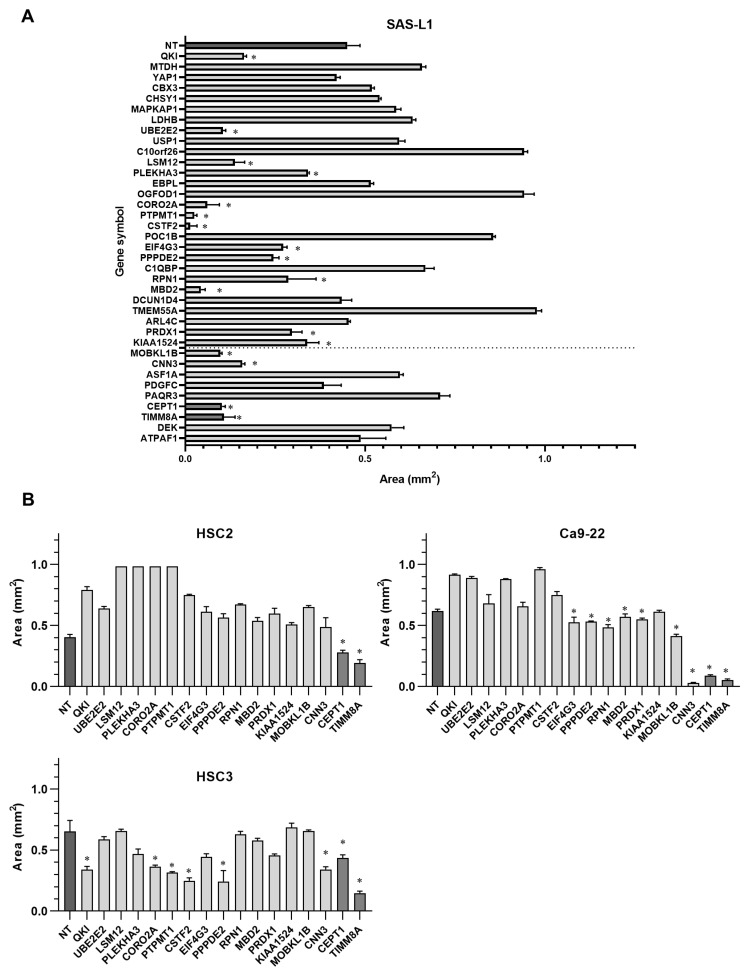
Identification of target genes involved in the migration inhibitory effect of miR-375-3p. (**A**) Cell migration was examined using siRNAs specific to miR-375-3p target genes. The knockdown of 16 genes significantly suppressed SAS-L1 cell migration. (**B**) CEPT1 and TIMM8A knockdown commonly inhibited the migration of all human OSCC cells. NT, non-targeting control siRNA; * *p* < 0.05 compared with the control; OSCC, oral squamous cell carcinoma; siRNA, small interfering RNA.

## Data Availability

The data supporting the reported results can be found at https://www.ncbi.nlm.nih.gov/geo/query/acc.cgi?acc=GSE240389, released on 13 August 2023, and https://www.ncbi.nlm.nih.gov/geo/query/acc.cgi?acc=GSE241289, released on 22 August 2023.

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
