# Peer review of "Possible Role of miR-375-3p in Cervical Lymph Node Metastasis of Oral Squamous Cell Carcinoma"

_cancers, 2024, doi:10.3390/cancers16081492_

Round 1

Reviewer 1 Report

Comments and Suggestions for Authors

In this manuscript, the authors identified the predictive role of miR-375-3p in cervical lymph node metastasis of OSCC. However, the experimental evidence and conclusions seem to lack sufficient data support. Below are some suggestions for improving the manuscript.

1. In the abstract, the emphasis should not be on detailing the experimental approach and steps; these sections should be expressed as succinctly as possible. The objectives, significance, results, and conclusions of the research are more crucial.

2. In Table S4, the microarray results for miRNA in primary OSCC tissues indicated significant differences in miR-375, 5580-3p, 617, 6746-5p, 451, 3935, etc. The rationale for choosing miR-375 and miR-6746-5p for subsequent experiments was unclear. What was justification for selecting these specific miRNAs?

3. In lines 240-241, the authors described differential results for miR-375-3p, while miR-6746-5p was not detected. However, Figure 1C only displayed results for miR-375 in N1-3, excluding miR-6746-5p. Moreover, the lack of differences in miR-6746-5p seems inconsistent with the miRNA microarray results.

4. In Figure 4A, the brightness of the miR-375-3P group's image was excessively high, obscuring the visualization of stained tumor cells. Please adjust the brightness and contrast appropriately to enhance the image's readability.

5. The Materials and Methods section did not mention RNA knockdown, and the supplementary materials did not provide siRNA sequences. This omission raised doubts about the claimed extensive knockdown work by the authors. Additionally, siRNA is prone to off-target effects, yet the authors have not detailed the quantity of each siRNA used or verified the knockdown efficiency. Addressing these concerns is essential for a more comprehensive understanding of the experimental procedures.

6. Relying solely on the results of CCK-8 and Transwell assays is insufficient to substantiate the use of miR-375-3p as a predictive marker for potential lymph node metastasis.  Its impact on cell motility, EMT, and migration-related proteins needs to be assessed.  Additionally, GSEA plots for downstream sequencing results of miR-375 should be presented.  Upon examining the authors' enriched downstream results, it appears that the role of miR-375 lacks significant correlation with any specific functions in cancer cell.

Comments on the Quality of English Language

The manuscript need to be improved but have no concern with language.

Author Response

Thank you very much for your valuable suggestions.

  1. In the abstract, the emphasis should not be on detailing the experimental approach and steps; these sections should be expressed as succinctly as possible. The objectives, significance, results, and conclusions of the research are more crucial.

Response: We have changed the text in the Abstract section as succinctly (line 20-32).

  1. In Table S4, the microarray results for miRNA in primary OSCC tissues indicated significant differences in miR-375, 5580-3p, 617, 6746-5p, 451, 3935, etc. The rationale for choosing miR-375 and miR-6746-5p for subsequent experiments was unclear. What was justification for selecting these specific miRNAs?

Response: In Table S4, we identified 82 miRNAs expressed in primary OSCC tissues with >2-fold changes compared to adjacent normal oral mucosa by miRNA microarray analysis. Among these 82 miRNAs, only the expression levels of miR-375-3p and miR-6746-5p showed a more than 2-fold decrease in primary OSCC tissues with latent cervical lymph node metastasis (line 252-255, Figure 1A and B).

  1. In lines 240-241, the authors described differential results for miR-375-3p, while miR-6746-5p was not detected. However, Figure 1C only displayed results for miR-375 in N1-3, excluding miR-6746-5p. Moreover, the lack of differences in miR-6746-5p seems inconsistent with the miRNA microarray results.

Response: We have added the expression level of miR-6746-5p by miRNA microarray analysis to the Figure1B. Subsequently, we performed RT-quantitative real-time PCR to confirm the expression of miR-6746-5p, but its expression could not be detected in almost all cases. We considered that the PCR amplification was not successful with commercially available TaqMan probe and primers (line 259-260).

  1. In Figure 4A, the brightness of the miR-375-3P group's image was excessively high, obscuring the visualization of stained tumor cells. Please adjust the brightness and contrast appropriately to enhance the image's readability.

Response: We have adjusted the brightness and contrast of Figure 4A.

  1. The Materials and Methods section did not mention RNA knockdown, and the supplementary materials did not provide siRNA sequences. This omission raised doubts about the claimed extensive knockdown work by the authors. Additionally, siRNA is prone to off-target effects, yet the authors have not detailed the quantity of each siRNA used or verified the knockdown efficiency. Addressing these concerns is essential for a more comprehensive understanding of the experimental procedures.

Response: Because there were a large number of target gene candidates (37 types), we used a very small amount of cherrypick siRNA library purchased from Horizon Discovery. Therefore, we did not evaluate their off-target effects and knockdown efficiencies. However, we used the siRNAs called ON-TARGETplus which has been chemically modified to suppress off-target effects. Furthermore, these products are a pool of 4 types of siRNA specific for each gene, but their sequences are not made public. The quantity of each siRNA was indicated in the Materials and Methods section (line 176-184).

  1. Relying solely on the results of CCK-8 and Transwell assays is insufficient to substantiate the use of miR-375-3p as a predictive marker for potential lymph node metastasis. Its impact on cell motility, EMT, and migration-related proteins needs to be assessed. Additionally, GSEA plots for downstream sequencing results of miR-375 should be presented. Upon examining the authors' enriched downstream results, it appears that the role of miR-375 lacks significant correlation with any specific functions in cancer cell.

Response: Invasion assay was also performed to assess the invasiveness of human OSCC cells, but the control group did not invade. We evaluated the effect of miR-375-3p mimic on the in vivo growth of human OSCC cells, and observed their inhibitory effect against OSCC tumor growth (Figure 5). We conducted the GSEA, but did not any changes in the expression of genes associated with lymph node metastasis (line 348-351, Table S7).

Reviewer 2 Report

Comments and Suggestions for Authors

Re: Possible role of miR-375-3p in cervical lymph node metastasis of oral squamous cell carcinoma  

This article entitled ‘Possible role of miR-375-3p in cervical lymph node metastasis of oral squamous cell carcinoma’ suggested that the down-regulation of miR-375-3p may enhance the growth and migration of OSCC cells by activating PI3K/AKT signal and up-regulating the expression of CEPT1 and TIMM8A-408, leading to cervical LNM in OSCC. The findings are interesting to facilitate the identification of potential biomarkers for the lymph node metastasis of OSCC. However, in order to achieve scientific publishing standards, there are still some issues should be addressed before. The detail comments and suggestions are followed.

1.      The main weakness of this study is lack of in vivo verification of miR-375-3p effect on lymph node metastasis of OSCC. This manuscript provided the potential relationship between miR-375-3p expression with LNM of OSCC in clinic samples, which did not support their hypothesis for the effect of miR-375-3p on LNM of OSCC, although the potential role of miR-375-3p on the mobility of OSCC cells. The animal models should be conducted to confirm the role of miR-375-3p on LNM of OSCC.

2.      In this study, the introduction explained “Treatment strategies for OSCC are determined based on clinical staging according to the TNM classification of the Union for International Cancer Control. Tumors less than 4 cm in size with a depth of invasion of less than 10 mm (T1/2) and no cervical lymph node metastasis (LNM) (N0) are classified as early OSCC (eOSCC, Stage I/II)”, while in Table S1, both stage I and stage II contain lymph node metastasis, so the clinical staging of OSCC should be further confirmed.

3.      The introduction part of the article lacks logic. The first paragraph of introduction puts forward that the treatment of OSCC is based on clinical stages, and only talks about early OSCC. The author should introduce the staging principle of OSCC, the treatment principle of different stages and the importance of lymph node prediction caused by the difficulty of treatment choice in more detail. This section should be re-written comprehensively.

4.      The key word of the article contains “microRNA(miRNA); microRNA-375-3p (miR-375); tumor suppressive-microRNA (TS-miR)”, which is repetitive. It is suggested that the author consider it again.

5.      In result 1 of Fig. 1A, after screening, the author suggested that the down-regulation of miR-375-3p and miR-6746-5p was related to the potential LNM, and then the experimental verification of miR-375-3p was carried out. Can you further explain why you chose miR-375-3p instead of miR-6746-5p?

6.      In Fig. 3 and Fig. 4, the function of cell proliferation and migration is verified after adding miR-375-3p mimic, and the transfection efficiency of miR-375-3p mimic should be verified before this. Similarly, as shown in Fig. 6 and Fig. 7, the siRNA transfection verification of the target gene also needs to be added.

7.      In Fig. 4A, the picture of cell migration is too vague, it is suggested to put a clearer original picture. In Transwell assy, migration and invasion should be both conducted to verify its mobility and invasiveness.

8.      In the Cell growth assay of materials and methods, the author used the method of counting cells first, then conducted the transfection experiment, and cell viability was detected 72 hours later. Does this experimental method take into account the effect of transfection reagents on cytotoxicity?

9.      The similar results of miR-375-3p mimic and knockdown of CEPT1/TIMM8A were not sufficient to support the downstream genes of miR-375-3p. Luciferase assay and/or RIP should be conducted for molecular mechanism analysis.

10.  In the discussion, on line 390, the author wrote "Detection of miR-375-3P in Saliva May be useful as a biomarker for potential malignant diseases of the oral cavity". Compared with normal tissues, the expression of miR-375-3P in OSCC without LNM is also very low (from the expression of miR-375-3P detected by the author in the cell line and Table S4). How to identify miR-375-3P as a biological indicator for predicting OSCC or LNM?

11.  In the conclusion, the author wrote " Downregulation of miR-375-3p enhanced the growth and migration of OSCC cells via the activation of PI3K/AKT signaling and upregulation of CEPT1 and TIMM8A expression and may cause cervical LNM in OSCC ". I can't accept this expression. The author only predicts that miR-375-3p may activate PI3K/AKT signal through data analysis, and there is no relevant experimental verification. In addition, the author knocked out the screened target gene of miR-375-3p to detect the cell proliferation and migration ability, which can only explain the influence of the target gene on the phenotype of OSCC cells, but can not directly explain the regulatory effect of miR-375-3p on the target gene. Moreover, CEPT1 and TIMM8A only affect the migration ability of OSCC cells, but have little effect on the proliferation ability of cells.

12.  There are mistakes in line 204 to 205 of the article. The author is advised to improve the article and grammar again. The English writing is poor. It is suggested to polish the manuscript by a native English speaker.

Re: Possible role of miR-375-3p in cervical lymph node metastasis of oral squamous cell carcinoma  

This article entitled ‘Possible role of miR-375-3p in cervical lymph node metastasis of oral squamous cell carcinoma’ suggested that the down-regulation of miR-375-3p may enhance the growth and migration of OSCC cells by activating PI3K/AKT signal and up-regulating the expression of CEPT1 and TIMM8A-408, leading to cervical LNM in OSCC. The findings are interesting to facilitate the identification of potential biomarkers for the lymph node metastasis of OSCC. However, in order to achieve scientific publishing standards, there are still some issues should be addressed before. The detail comments and suggestions are followed.

1.      The main weakness of this study is lack of in vivo verification of miR-375-3p effect on lymph node metastasis of OSCC. This manuscript provided the potential relationship between miR-375-3p expression with LNM of OSCC in clinic samples, which did not support their hypothesis for the effect of miR-375-3p on LNM of OSCC, although the potential role of miR-375-3p on the mobility of OSCC cells. The animal models should be conducted to confirm the role of miR-375-3p on LNM of OSCC.

2.      In this study, the introduction explained “Treatment strategies for OSCC are determined based on clinical staging according to the TNM classification of the Union for International Cancer Control. Tumors less than 4 cm in size with a depth of invasion of less than 10 mm (T1/2) and no cervical lymph node metastasis (LNM) (N0) are classified as early OSCC (eOSCC, Stage I/II)”, while in Table S1, both stage I and stage II contain lymph node metastasis, so the clinical staging of OSCC should be further confirmed.

3.      The introduction part of the article lacks logic. The first paragraph of introduction puts forward that the treatment of OSCC is based on clinical stages, and only talks about early OSCC. The author should introduce the staging principle of OSCC, the treatment principle of different stages and the importance of lymph node prediction caused by the difficulty of treatment choice in more detail. This section should be re-written comprehensively.

4.      The key word of the article contains “microRNA(miRNA); microRNA-375-3p (miR-375); tumor suppressive-microRNA (TS-miR)”, which is repetitive. It is suggested that the author consider it again.

5.      In result 1 of Fig. 1A, after screening, the author suggested that the down-regulation of miR-375-3p and miR-6746-5p was related to the potential LNM, and then the experimental verification of miR-375-3p was carried out. Can you further explain why you chose miR-375-3p instead of miR-6746-5p?

6.      In Fig. 3 and Fig. 4, the function of cell proliferation and migration is verified after adding miR-375-3p mimic, and the transfection efficiency of miR-375-3p mimic should be verified before this. Similarly, as shown in Fig. 6 and Fig. 7, the siRNA transfection verification of the target gene also needs to be added.

7.      In Fig. 4A, the picture of cell migration is too vague, it is suggested to put a clearer original picture. In Transwell assy, migration and invasion should be both conducted to verify its mobility and invasiveness.

8.      In the Cell growth assay of materials and methods, the author used the method of counting cells first, then conducted the transfection experiment, and cell viability was detected 72 hours later. Does this experimental method take into account the effect of transfection reagents on cytotoxicity?

9.      The similar results of miR-375-3p mimic and knockdown of CEPT1/TIMM8A were not sufficient to support the downstream genes of miR-375-3p. Luciferase assay and/or RIP should be conducted for molecular mechanism analysis.

10.  In the discussion, on line 390, the author wrote "Detection of miR-375-3P in Saliva May be useful as a biomarker for potential malignant diseases of the oral cavity". Compared with normal tissues, the expression of miR-375-3P in OSCC without LNM is also very low (from the expression of miR-375-3P detected by the author in the cell line and Table S4). How to identify miR-375-3P as a biological indicator for predicting OSCC or LNM?

11.  In the conclusion, the author wrote " Downregulation of miR-375-3p enhanced the growth and migration of OSCC cells via the activation of PI3K/AKT signaling and upregulation of CEPT1 and TIMM8A expression and may cause cervical LNM in OSCC ". I can't accept this expression. The author only predicts that miR-375-3p may activate PI3K/AKT signal through data analysis, and there is no relevant experimental verification. In addition, the author knocked out the screened target gene of miR-375-3p to detect the cell proliferation and migration ability, which can only explain the influence of the target gene on the phenotype of OSCC cells, but can not directly explain the regulatory effect of miR-375-3p on the target gene. Moreover, CEPT1 and TIMM8A only affect the migration ability of OSCC cells, but have little effect on the proliferation ability of cells.

12.  There are mistakes in line 204 to 205 of the article. The author is advised to improve the article and grammar again. The English writing is poor. It is suggested to polish the manuscript by a native English speaker.

Comments on the Quality of English Language

Re: Possible role of miR-375-3p in cervical lymph node metastasis of oral squamous cell carcinoma  

This article entitled ‘Possible role of miR-375-3p in cervical lymph node metastasis of oral squamous cell carcinoma’ suggested that the down-regulation of miR-375-3p may enhance the growth and migration of OSCC cells by activating PI3K/AKT signal and up-regulating the expression of CEPT1 and TIMM8A-408, leading to cervical LNM in OSCC. The findings are interesting to facilitate the identification of potential biomarkers for the lymph node metastasis of OSCC. However, in order to achieve scientific publishing standards, there are still some issues should be addressed before. The detail comments and suggestions are followed.

1.      The main weakness of this study is lack of in vivo verification of miR-375-3p effect on lymph node metastasis of OSCC. This manuscript provided the potential relationship between miR-375-3p expression with LNM of OSCC in clinic samples, which did not support their hypothesis for the effect of miR-375-3p on LNM of OSCC, although the potential role of miR-375-3p on the mobility of OSCC cells. The animal models should be conducted to confirm the role of miR-375-3p on LNM of OSCC.

2.      In this study, the introduction explained “Treatment strategies for OSCC are determined based on clinical staging according to the TNM classification of the Union for International Cancer Control. Tumors less than 4 cm in size with a depth of invasion of less than 10 mm (T1/2) and no cervical lymph node metastasis (LNM) (N0) are classified as early OSCC (eOSCC, Stage I/II)”, while in Table S1, both stage I and stage II contain lymph node metastasis, so the clinical staging of OSCC should be further confirmed.

3.      The introduction part of the article lacks logic. The first paragraph of introduction puts forward that the treatment of OSCC is based on clinical stages, and only talks about early OSCC. The author should introduce the staging principle of OSCC, the treatment principle of different stages and the importance of lymph node prediction caused by the difficulty of treatment choice in more detail. This section should be re-written comprehensively.

4.      The key word of the article contains “microRNA(miRNA); microRNA-375-3p (miR-375); tumor suppressive-microRNA (TS-miR)”, which is repetitive. It is suggested that the author consider it again.

5.      In result 1 of Fig. 1A, after screening, the author suggested that the down-regulation of miR-375-3p and miR-6746-5p was related to the potential LNM, and then the experimental verification of miR-375-3p was carried out. Can you further explain why you chose miR-375-3p instead of miR-6746-5p?

6.      In Fig. 3 and Fig. 4, the function of cell proliferation and migration is verified after adding miR-375-3p mimic, and the transfection efficiency of miR-375-3p mimic should be verified before this. Similarly, as shown in Fig. 6 and Fig. 7, the siRNA transfection verification of the target gene also needs to be added.

7.      In Fig. 4A, the picture of cell migration is too vague, it is suggested to put a clearer original picture. In Transwell assy, migration and invasion should be both conducted to verify its mobility and invasiveness.

8.      In the Cell growth assay of materials and methods, the author used the method of counting cells first, then conducted the transfection experiment, and cell viability was detected 72 hours later. Does this experimental method take into account the effect of transfection reagents on cytotoxicity?

9.      The similar results of miR-375-3p mimic and knockdown of CEPT1/TIMM8A were not sufficient to support the downstream genes of miR-375-3p. Luciferase assay and/or RIP should be conducted for molecular mechanism analysis.

10.  In the discussion, on line 390, the author wrote "Detection of miR-375-3P in Saliva May be useful as a biomarker for potential malignant diseases of the oral cavity". Compared with normal tissues, the expression of miR-375-3P in OSCC without LNM is also very low (from the expression of miR-375-3P detected by the author in the cell line and Table S4). How to identify miR-375-3P as a biological indicator for predicting OSCC or LNM?

11.  In the conclusion, the author wrote " Downregulation of miR-375-3p enhanced the growth and migration of OSCC cells via the activation of PI3K/AKT signaling and upregulation of CEPT1 and TIMM8A expression and may cause cervical LNM in OSCC ". I can't accept this expression. The author only predicts that miR-375-3p may activate PI3K/AKT signal through data analysis, and there is no relevant experimental verification. In addition, the author knocked out the screened target gene of miR-375-3p to detect the cell proliferation and migration ability, which can only explain the influence of the target gene on the phenotype of OSCC cells, but can not directly explain the regulatory effect of miR-375-3p on the target gene. Moreover, CEPT1 and TIMM8A only affect the migration ability of OSCC cells, but have little effect on the proliferation ability of cells.

12.  There are mistakes in line 204 to 205 of the article. The author is advised to improve the article and grammar again. The English writing is poor. It is suggested to polish the manuscript by a native English speaker.

Author Response

Thank you very much for your valuable suggestions.

  1. The main weakness of this study is lack of in vivo verification of miR-375-3p effect on lymph node metastasis of OSCC. This manuscript provided the potential relationship between miR-375-3p expression with LNM of OSCC in clinic samples, which did not support their hypothesis for the effect of miR-375-3p on LNM of OSCC, although the potential role of miR-375-3p on the mobility of OSCC cells. The animal models should be conducted to confirm the role of miR-375-3p on LNM of OSCC.

Response: There have been some reports of evaluating lymph node metastasis by introducing tumors into the tongues of nude mice, but unfortunately it was difficult to evaluate small lymph node metastases. Therefore, we used a commonly practiced model of creating subcutaneous tumors in the blank to verify in vivo tumor growth involved in lymph node metastasis (line 454-464, Figure 5).

  1. In this study, the introduction explained “Treatment strategies for OSCC are determined based on clinical staging according to the TNM classification of the Union for International Cancer Control. Tumors less than 4 cm in size with a depth of invasion of less than 10 mm (T1/2) and no cervical lymph node metastasis (LNM) (N0) are classified as early OSCC (eOSCC, Stage I/II)”, while in Table S1, both stage I and stage II contain lymph node metastasis, so the clinical staging of OSCC should be further confirmed.

Response: All in this study were clinical Stage I/II oral squamous cell carcinoma cases. Among these, patients who developed late cervical lymph node metastasis during the follow-up period and patients whose sentinel lymph node biopsy was positive in the primary surgery were classified as the LNM-positive group. We have changed the text in the Table S1.

  1. The introduction part of the article lacks logic. The first paragraph of introduction puts forward that the treatment of OSCC is based on clinical stages, and only talks about early OSCC. The author should introduce the staging principle of OSCC, the treatment principle of different stages and the importance of lymph node prediction caused by the difficulty of treatment choice in more detail. This section should be re-written comprehensively.

Response: We have added the treatments for advanced OSCC (line 44-49).

  1. The key word of the article contains “microRNA(miRNA); microRNA-375-3p (miR-375); tumor suppressive-microRNA (TS-miR)”, which is repetitive. It is suggested that the author consider it again.

Response: The keyword "microRNA (miRNA)" has been deleted (line 33-34).

  1. In result 1 of Fig. 1A, after screening, the author suggested that the down-regulation of miR-375-3p and miR-6746-5p was related to the potential LNM, and then the experimental verification of miR-375-3p was carried out. Can you further explain why you chose miR-375-3p instead of miR-6746-5p?

Response: We have added the expression level of miR-6746-5p by miRNA microarray analysis to the Figure1B. Subsequently, we performed RT-quantitative real-time PCR to confirm the expression of miR-6746-5p, but its expression could not be detected in almost all cases. We considered that the PCR amplification was not successful with commercially available TaqMan probe and primers (line 259-260).

  1. In Fig. 3 and Fig. 4, the function of cell proliferation and migration is verified after adding miR-375-3p mimic, and the transfection efficiency of miR-375-3p mimic should be verified before this. Similarly, as shown in Fig. 6 and Fig. 7, the siRNA transfection verification of the target gene also needs to be added.

Response: The overexpression of miR-375 after its mimic introduction was verified by RT-qPCR. For siRNA, we select libraries with high introduction efficiency. Because there were a large number of target gene candidates (37 types), we used a very small amount of cherrypick siRNA library purchased from Horizon Discovery. Therefore, we did not evaluate their knockdown efficiencies. Furthermore, these products are a pool of 4 types of siRNA specific for each gene and guaranteed their knockdown effects.

  1. In Fig. 4A, the picture of cell migration is too vague, it is suggested to put a clearer original picture. In Transwell assy, migration and invasion should be both conducted to verify its mobility and invasiveness.

Response: The images has been changed (Figure4A). Invasion assay was also performed to assess the invasiveness of human OSCC cells, but the control group did not invade.

  1. In the Cell growth assay of materials and methods, the author used the method of counting cells first, then conducted the transfection experiment, and cell viability was detected 72 hours later. Does this experimental method take into account the effect of transfection reagents on cytotoxicity?

Response: Our laboratory has previously tested the transfection conditions of miRNA and siRNA that are similar to the reagents used in this research, and we confirmed that 20 nM for miRNA mimics, 10 nM for siRNAs, and 0.3% Lipofectamine RNAiMAX are not cytotoxic.

  1. The similar results of miR-375-3p mimic and knockdown of CEPT1/TIMM8A were not sufficient to support the downstream genes of miR-375-3p. Luciferase assay and/or RIP should be conducted for molecular mechanism analysis.

Response: RIP and luciferase assays are on going. We plan to indicate these results in our next paper (line 451-453).

  1. In the discussion, on line 390, the author wrote "Detection of miR-375-3P in Saliva May be useful as a biomarker for potential malignant diseases of the oral cavity". Compared with normal tissues, the expression of miR-375-3P in OSCC without LNM is also very low (from the expression of miR-375-3P detected by the author in the cell line and Table S4). How to identify miR-375-3P as a biological indicator for predicting OSCC or LNM?

Response: We consider that low expression of miR-375 can be quantified as a biological indicator by performing absolute quantification using a highly sensitive digital PCR device.

  1. In the conclusion, the author wrote " Downregulation of miR-375-3p enhanced the growth and migration of OSCC cells via the activation of PI3K/AKT signaling and upregulation of CEPT1 and TIMM8A expression and may cause cervical LNM in OSCC ". I can't accept this expression. The author only predicts that miR-375-3p may activate PI3K/AKT signal through data analysis, and there is no relevant experimental verification. In addition, the author knocked out the screened target gene of miR-375-3p to detect the cell proliferation and migration ability, which can only explain the influence of the target gene on the phenotype of OSCC cells, but cannot directly explain the regulatory effect of miR-375-3p on the target gene. Moreover, CEPT1 and TIMM8A only affect the migration ability of OSCC cells, but have little effect on the proliferation ability of cells.

Response: We have changed the conclusions (line 466-471).

  1. There are mistakes in line 204 to 205 of the article. The author is advised to improve the article and grammar again. The English writing is poor. It is suggested to polish the manuscript by a native English speaker.

Response: English language editing was performed by Editage (www.editage.jp).

Reviewer 3 Report

Comments and Suggestions for Authors

Thank you for allowing me to review this in vitro study aimed to explore novel biomarkers for predicting latent cervical LNM and examine their functions by quantifying the expression of miRNAs involved in migration and proliferation.

The introduction, methodology and results are adequately written, and I congratulate the authors.

The discussion is somewhat scarce and does not sufficiently explain the results. I definitely recommend the authors to add the strength of the study, its limitations and recommendations for future research.

The literature is listed according to the instructions of the Journal.

Author Response

The discussion is somewhat scarce and does not sufficiently explain the results. I definitely recommend the authors to add the strength of the study, its limitations and recommendations for future research.

Response : Thank you very much for your valuable suggestions. We have changed the discussion section (line 451-464).

Reviewer 4 Report

Comments and Suggestions for Authors

In the present study, the authors  showed that the downregulation of miR-375-3p expression was associated with latent cervical LNM of eOSCC, and the replenishment of miR-375-3p suppressed the growth and migration of human OSCC cells.

Downregulation of miR-375-3p enhanced the growth and migration of OSCC cells  via the activation of PI3K/AKT signaling and upregulation of CEPT1 and TIMM8A ex- pression and may cause cervical LNM in OSCC. The expression levels of miR-375-3p in primary tumor tissues appear to be a useful biomarker for predicting latent cervical LNM  in eOSCC

The overall manuscript structure, including sections and its subheadings, text flow, and writing are appropriate."

The title should include the study design, which is an in vitro study.

the objectives of the study are clearly stated in the manuscript.

The in vitro study was adequate, and tests were conducted in triplicate."Microarray and pathway analyses should be explained more extensively.

Statistical analyses, controls, sampling mechanism, and statistical reporting  are appropriate and well described.

Figure 5B is difficult to understand, and Table 1 should be supplementary.

The quality of figures 6 and 7 should be improved.

The interpretation of results and study conclusions are supported by the data and the study design

Please discuss any unexpected findings and their potential implications.In the discussion section, please clearly emphasize the strengths of your study/theory/methods/argument. Suggestions for improving the discussion of limitations

Author Response

Thank you for valuable suggestion.

1.The title should include the study design, which is an in vitro study.

Response: Since in vivo study was newly added (Figure 5), the title was not changed.

2.The objectives of the study are clearly stated in the manuscript.

Response: We described in the Introduction section (line 102-104).

3.The in vitro study was adequate, and tests were conducted in triplicate. "Microarray and pathway analyses should be explained more extensively.

Response: We described in the Materials and Methods section (line 233-236).

4.Figure 5B is difficult to understand, and Table 1 should be supplementary.

Response: We have improved the resolution of Figure 5B to make it easier to see (Figure 6B). Table 1 has been moved to supplementary (Table S6).

5.The quality of figures 6 and 7 should be improved.

Response: We improved the quality of the images (Figure 7 and 8).

6.Please discuss any unexpected findings and their potential implications. In the discussion section, please clearly emphasize the strengths of your study/theory/methods/argument. Suggestions for improving the discussion of limitations.

Response : We described in the Discussion section (line 451-464).

Round 2

Reviewer 4 Report

Comments and Suggestions for Authors

I agree with the changes